# The Incidence of Falls and Related Factors among Chinese Elderly Community Residents in Six Provinces

**DOI:** 10.3390/ijerph192214843

**Published:** 2022-11-11

**Authors:** Kun Wang, Meijun Chen, Xiaoyue Zhang, Lanchao Zhang, Chun Chang, Yu Tian, Xiaofeng Wang, Zhijing Li, Ying Ji

**Affiliations:** 1Department of Social Medicine and Health Education, School of Public Health, Peking University Health Science Center, Beijing 100191, China; 2School of Public Health, The University of Hong Kong, Hong Kong 999077, China; 3Xinjiekou Community Health Service Center of Xicheng District, Beijing 100035, China; 4Ronghua Community Health Service Center of Beijing Economic-Technological Development Area, Beijing 100176, China; 5School of Health Humanities, Peking University Health Science Center, Beijing 100191, China

**Keywords:** falls, Chinese older people, risk factors, self-health management awareness, family support

## Abstract

This cross-sectional study classified the factors related to falls among residents ≥ 60 years old in China in order to provide evidence for the prevention of falls in the elderly. A total of 2994 participants were enrolled, and the correlations between social demography, physical health, self-health management awareness, family support, and fall risk were analyzed. Factors influencing falls were classified by location, cause, and the activity during falls. Suffering from osteoarthropathy (OR = 1.761, 95% CI: 1.234–2.513, *p* < 0.05), depression or anxiety (OR = 1.896, 95% CI: 1.331–2.700, *p* < 0.001), household size > 2 (OR = 1.806, 95% CI: 1.042–3.130, *p* < 0.05), and poor self-assessed health (OR = 1.478, 95% CI: 1.107–1.972, *p* < 0.01) were risk factors. Higher participation in community health programs (OR = 0.522, 95% CI: 0.298–0.912, *p* < 0.05) and spousal support (OR = 0.909, 95% CI: 0.841–0.981, *p* < 0.05) were protective factors. Falls were divided into the following categories: stairs/hallway (vision, attention problems), bath/toilet (vision, attention problems, slipping), indoor housework (dizziness, leg weakness), and outdoor activities (attention, surface problems). While acknowledging that the personal physical and mental health of the elderly may lead to falls, community support, accelerated transformation of housing, and the construction of barrier-free environments play an important role in reducing the risk of falls.

## 1. Introduction

Falls are a leading cause of injury and death among older adults and a serious public health problem. According to the WHO, falls are the second leading cause of unintentional injury deaths globally, with 684,000 people dying from falls each year, many of whom are over 60 years old [1]. The prevalence of fall risk factors sharply increases after age 70 [2], and about 32–42% of people over 70 years old fall each year [3]. As the fall risk increases, the rates of hospitalization and mortality after a fall also increase [4]. This seriously affects the physical and mental health of the elderly, causing problems such as ataxia, fear of falling again, and social isolation [5,6], thereby affecting their quality of life, and introducing serious care and economic burdens to families [7].

The world’s population is aging [8], with developing countries aging the fastest [9]. Compared with those in developed countries, elderly people in developing countries have significantly higher rates of falls [10]. The socio-environmental background of developing countries also puts older adults at a higher risk of falls, including high-risk living environments, misconceptions that falls are inevitable among older adults, and inadequate health care services [11]. Problems in the living environment faced by the elderly in developing countries need to be solved urgently. Specifically, awareness regarding fall prevention should be improved, and the role of community support should be actively played. For developing countries, the effectiveness of national fall prevention guidelines needs to be verified [11] and should be supplemented and adjusted in response to real-life situations.

Falls are often caused by a variety of factors, including personal and environmental. The most common personal factors [12] are physical disorders, such as visual impairment, musculoskeletal problems, and chronic diseases [13,14]. In addition, personal psychological status is also closely related to falls [15,16]. Health-related behaviors, such as physical activity and diet, are also closely related to falls [17,18]. Self-management awareness to prevent falls is generally lacking in older adults [19]. The improvement of self-health management awareness can promote the development of healthy behaviors and increase the utilization of community resources, which will lead to the better prevention of falls [20]. However, there is still a lack of research on the correlation between self-health management awareness and falls. Social environmental factors are also closely related to falls in the elderly. In recent years, more attention has been paid to the relationship between family support, including emotional support, family living situation, and the health of the elderly population [21,22,23,24]. However, there are few studies linking family support with falls in older adults. In addition, in previous studies, there has been no direct conclusion on environmental exposure as a risk factor and the occurrence of falls because the environmental factors of falls are complicated to assess [25,26,27]. Therefore, in addition to the common personal factors, we also focus on the self-health management awareness, family support, and environmental exposure, which are also important for falls in older adults in recent years but are less well researched.

China is one of the countries with the largest fall-related disease burden in the world [7], and falls are the leading cause of injury-related death among Chinese aged 65 and older [28]. Therefore, it is of great practical significance to conduct research on the factors influencing falls in the elderly in China. The social environment and customs in China differ from those in Western countries. The elderly in China are more dependent on their children both materially and emotionally, and it is common for them to live with their children [29]. Thus, family support plays a key role in later life. In addition, the elderly in China lack awareness on preventing falls. Chinese families with low socioeconomic status seldom implement housing renovation suitable for aging, and there are relatively few architectural designs for fall prevention in rural areas. There is still a lack of empirical analysis and research on fall scenarios [30]. Furthermore, most research data for the Chinese elderly population are from the China Health and Retirement Longitudinal Study (CHARLS) database [15,31], limited to single variables. In addition, many studies have a small sample size [32], and the research participants are limited to a certain area of China [33,34], which is not representative. Therefore, we carried out the large-scale research representative of China to enrich the theoretical evidence.

In order to reduce falls among the elderly, the Chinese government has adopted a series of policy measures. “Healthy China Action (2019–2030)” indicated that “emphasis on muscle strength and flexibility exercises, proper balance exercises, strong musculoskeletal system, and prevention of falls” is needed. The National Disability Prevention Action Plan (2021–2025) also stressed that it is necessary to “strengthen the prevention and control of disability due to falls in the elderly”, calling on the elderly to be the first responsible person for preventing falls [35,36]. In addition, the nationally basic public health service project was officially implemented in 2009, which established and improved the health service system for the elderly in China. The elderly now have the right to receive injury prevention, such as that from falls, and health education [37]. Numerous regulations empower older adults to avoid falls. Therefore, it is necessary to understand the effect of social health services received by the elderly in China, which will also provide avenues and guidance for the prevention of falls in the elderly in developing countries and the world in the future.

In this study, the target population was aged 60 years and older. In the context of the particularity of the families of the elderly in China and improving the provision of basic health services, this study describes the occurrence and characteristics of falls among the elderly in China and explores the related factors in multiple aspects, including sociodemographic characteristics, physical health, self-health management awareness, and family support. We also explore the role of environmental factors among the elderly associated with different types of falls to provide more evidence for the prevention of falls, aiming at monitoring, intervening, educating high-risk groups, and proposing recommendations.

## 2. Materials and Methods

### 2.1. Participants

The participants of the study were people of 60 years old and over. By adopting a stratified multi-stage random sampling method, we randomly selected 2 provinces in the eastern, central, and western regions in China, and in each province, the provincial capital city and a city whose GDP ranks in the bottom 1/3 of the province were selected. In each city, 1 district (urban area) and 1 county (rural area) were selected according to the urban and rural stratification, and 1 street (township) was chosen from each district (county). Details are shown in Figure 1. Then, in the smallest administrative area unit, community (administrative village), the respondents were selected according to the household registration list, and a total of 3004 elderly people were included from 6 provinces. For the purpose of this study, 10 respondents lacking information about falls were excluded. In the end, there were 2994 valid questionnaires, with an effective return rate of 99.7%. This study was approved by the Biomedical Ethics Committee of Peking University (IRB00001052-19143).

### 2.2. Instrument

A self-designed questionnaire was used to conduct the survey. Based on previous research, the questionnaire included four dimensions: sociodemographic characteristics, physical health, self-health management awareness, and family support. A combination of household and centralized surveys was adopted. For the elderly with mobility, the community (administrative village) organized a centralized survey, and for those with difficulty in moving, household surveys were conducted. Furthermore, the respondents completed the questionnaires themselves. If the respondents could not complete the questionnaire independently, face-to-face interviews were conducted. The consent and cooperation of the respondents were obtained before the investigation. The entire investigation had a strict quality control and verification system and was conducted in accordance with the Declaration of Helsinki.

### 2.3. Research Content and Variable Definitions

#### 2.3.1. Falls

A fall was defined as an unintentional fall to the ground or a lower level, not due to an acute event (such as a seizure, syncope, or stroke) or overwhelming force [2]. The American and British Geriatrics Societies clinical practice guideline recommended that all elderly people should be asked about any falls that occurred in the previous year. In this study, respondents were asked to report whether they had fallen and the number of falls within the past 1 year; if fall(s) occurred, they were asked to report the specific circumstances of the latest fall (type, location, activity during the fall, cause, injury).

#### 2.3.2. Sociodemographic Characteristics

The participants’ gender, age, household registration, educational level, and monthly income were collected.

#### 2.3.3. Physical Health

Physical health assessment included prevalence of chronic illness, self- assessed health, and anxiety or depression. Firstly, the respondents reported chronic diseases that had been diagnosed by a doctor, including hypertension, diabetes, hyperlipidemia, osteoarthritis, cerebrovascular disease, coronary heart disease, chronic obstructive pulmonary disease, tumors, and other chronic diseases. Since diseases of the musculoskeletal system are closely related to falls, which can lead to weak muscles, unsteady gait, and poor balance, which are risk factors for falls [2]. In this study, chronic diseases were divided into osteoarthritis and other chronic diseases except osteoarthritis. Secondly, self-assessed health status was collected from the respondents on a 0–100 visual analogue scale (from poor to good) according to their health status on the day of the survey. In this study, the median was selected as the cut-off value, and scores ≤ 80 points were defined as poor health, and scores > 80 points were defined as good health. Thirdly, anxiety or depression was self-reported by asking whether they had feelings of depression or anxiety prior. (“Please point out your current physical condition based on the actual situation.” Responses included “No anxiety/depression” or “anxiety/depression”).

#### 2.3.4. Self-health Management Awareness

Self-health management awareness included health knowledge, regular physical activity, and the utilization of basic community health service programs. Self-health management refers to choosing appropriate methods to make optimal decisions and behaviors and improve health behaviors and outcomes through the acquired health knowledge, which includes planning for behaviors such as physical activity [38]. This study assumed that the elderly with better health knowledge and self-health management actions have better health management awareness.

According to the Department of Aging Health, National Health Commission of China, the health knowledge score was determined by asking the respondents general health questions (i.e., “Have you known the best energy distribution ratio of three meals a day?”. Four option including “breakfast 30%, lunch 40%, dinner 30%”, “breakfast 40%, lunch 40%, dinner 20%”, “breakfast 30%, lunch 50%, dinner 20%”, and “I don’t know”; “Have you known the best daily exercise time?”. Four options, including “6–7 a.m. and 3–5 p.m.”, “10–11 a.m. and 3–5 p.m.”, “10–11 a.m. and 6–7 p.m.”, and “I don’t know”), with 1 point for every correct answer, and a total score of 0–20 points. The higher the score, the better the level of health knowledge.

Regular physical activity in this study was defined as non-work and housework activities, including but not limited to walking, running, swimming, square dancing, cycling, etc. The respondents self-reported whether they undertook regular physical activity in daily life (“For the last six months, do you have regular physical activity?”. Responses included yes or no).

The social basic health service utilization in this study referred to whether the respondents participated in the three services of “free health checkup”, “contracted family doctor” and “health education” in the past year (“Have you received free health checkup from community health services in the past year?”, “Have you received contracted family doctor service from community health services in the past year?”, and “Have you received any health education from community health services in the past year?” Each response included yes or no. The participants were divided into four groups according to the type of service programs received: “none”, “1 program”, “2 programs”, and “3 programs”).

#### 2.3.5. Family Support

Household size and daily support provided by spouses were used as measures of family support. The number of family members measures the potential support of family members living together [39]. (“What is your family size?”. The participants were divided into three groups according to family size: “live alone”, “2 people”, and “>2 people”). Questions from the Xiao Shuiyuan Social Support Scale were used to determine whether the family receives support from the spouse [40], including certain economic support and help to solve practical problems and comfort and concern of the spouse, with a range of 1–6. (“Have you received support or care from your spouse?”. Responses comprised four options, including “never”, “seldom”, “usually”, and “always”; “In the past, have you received financial support and practical help from your spouse?”. Responses included yes or no. “In the past, have you received financial support and help in solving practical problems from your spouse?”. Responses included yes or no). The higher the score, the better the spousal support.

### 2.4. Data Analysis

SPSS 26.0 (IBM, Armonk, NY, USA) was used to analyze the data. For descriptive statistical analysis, frequency and percentage were used for count data, and mean ± standard deviation for measurement data. Chi-square and t tests were used for univariate analysis and the binary logistic regression analysis of categorical and continuous variables, respectively. Binary logistics regression was used to screen the characteristic parameters by the backward method, and the screening standard was 0.05; the likelihood ratio test was used as the test method, and the test level was 0.05.

The potential category analysis of fall descriptors was conducted using Mplus 8.3 (Los Angeles, CA, USA: Muthén and Muthén). The Lo–Mendel–Rubin likelihood ratio test (LMR-LRT), bootstrap likelihood ratio test (BLRT), Akaike information criterion (AIC), Bayesian information criterion (BIC), and entropy were used to screen characteristic models of falls occurrence. Among them, a *p* value of LMR-LRT and BLRT index < 0.05 indicated that the model fit well; the lower the values of AIC and BIC, the better the model fit. Entropy represents the probability of correct classification, and the higher the value, the better the model. Finally, we synthesized the above indicators and selected the optimal model.

## 3. Results

### 3.1. Descriptive Analysis

A total of 2994 elderly people were included as follows: 496 in Beijing (16.6%), 498 in Gansu (16.6%), 496 in Jiangxi (16.6%), 499 in Liaoning (16.7%), 489 in Zhejiang (16.3%), and 516 in Chongqing (17.2%). There were 1314 males (44.0%) and 1673 females (56.0%). There were 1774 participants under 70 years old (59.4%) and 1215 participants aged 70 years old or above (40.6%). There were 1475 participants (49.3%) with urban household registration and 1516 participants (50.7%) with rural household registration. There were 491 participants (16.4%) who were illiterate, 970 (32.4%) with primary education, 891 (29.8%) with junior high school education, and 642 (21.4%) with high school education and above. There were 795 participants (26.7%) with a monthly income less than CNY 500, 1400 (46.9%) with an income of CNY 500–2000, and 787 (26.4%) with an income of more than CNY 2000. The details are shown in Table 1.

### 3.2. Univariate Analysis of Falls and Injuries in the Elderly

There were significant differences in falls depending on whether participants had osteoarthrosis, anxiety/depression, participation in community health service, self-assessed health status, and spousal support (*p* < 0.05), as shown in Table 1.

### 3.3. Multivariate Analysis of Falls in the Elderly

Using the no fall group as the reference and including all variables in univariate analysis, the results of binary logistics regression revealed that suffering from osteoarthritis (OR = 1.761, 95% CI: 1.234–2.513, *p* < 0.05), depression or anxiety (OR = 1.896, 95% CI: 1.331–2.700, *p* < 0.001), poor self-assessed health (OR = 1.478, 95% CI: 1.107–1.972, *p* < 0.01), and family size > 2 people (OR = 1.806, 95% CI: 1.042–3.130, *p* < 0.05) were risk factors for falls. Participation in three community health service programs (OR = 0.522, 95% CI: 0.298–0.912, *p* < 0.05) and spousal support (OR = 0.909, 95% CI: 0.841–0.981, *p* < 0.05) were protective factors against falls. The details are shown in Table 2.

### 3.4. Behavioral Characteristics of Falls in the Elderly

The fall rate among the elderly in this study was 8.7%. Slips and trips accounted for 28.3% and 18.9%, respectively. The main places of falls were outdoors; bedroom, living room, or balcony; and bathroom or toilet; accounting for 40.6%, 18.0%, and 14.1%, respectively. Participants were prone to falling when walking or relaxing and doing housework, accounting for 38.6% and 18.1%. The most common causes of falls were a lack of concentration and uneven or slippery surfaces, accounting for 31.1% and 25.5%, respectively. Only 29.7% of the elderly who had fallen were not injured. Detailed information is shown in Table 3.

### 3.5. Analysis of Potential Categories of Falls in the Elderly

Fifteen factors including the location of the fall, activity engaged in when falling, and the causes of the fall were taken as explicit variables, which were divided into 1–5 categories for model fitting estimation. The results indicated that under the premise of meeting the LMR-LRT and BLRT tests, the four-category model had the smallest AIC and BIC values, the entropy values were larger, and the sample composition was relatively balanced. Compared with three categories, the conditional probability differences among four categories were larger, and therefore, the four-category model was chosen as the best model. See Table 4.

The elderly in Category 1 had the highest probability of falling on stairs/hallway and were mainly engaged in leisure activities when they fell. The causes of the fall were characterized by poor eyesight and inattention, so Category 1 was named “stairs/hallways–vision and attention problem fall type”, accounting for 27.2%. The elderly in Category 2 were characterized by poor eyesight, lack of concentration, or an uneven/slippery surface when they were in the bathroom/toilet to take a bath/go to the toilet, so Category 2 was named “bath/toilet–vision and attention problem slip type”, accounting for 10.5%. Category 3 had a high probability of falling in the bedroom/balcony and kitchen, and the main activity when falling was housework, mainly due to dizziness and weak legs, so Category 3 was named “indoor housework–dizziness and leg weakness fall type”, accounting for 27.6%. The elderly in Category 4 mainly fell outdoors and were engaged in leisure activities, and they fell due to lack of concentration or uneven or slippery surfaces; therefore, Category 4 was named “outdoor activities–attention and surface problem fall type”, accounting for 34.7%. Details are shown in Figure 2.

## 4. Discussion

This study investigated the falls of 2994 Chinese elderly people over 60 years old and analyzed the influencing factors of falls from four dimensions: social demographic characteristics, physical health, self-health management awareness, and family support. The results showed that suffering from osteoarthrosis, depression or anxiety, self-assessed physical health status, participating in several community health service programs, the number of family members, and spousal support in terms of family support were related to falls among the elderly. Among the people who fell, most slipped or tripped, and most falls occurred outdoors. The activities taking place when they fell were mainly walking/relaxing, and most fell because of inattention and surface problems. Most participants (70.3%) were injured by the fall. In addition, according to the location of the fall, the activity occurring when falling, and the cause of the fall, the falls were divided into four types: stairs/hallways—vision and attention problem fall type; bath/toilet—vision and attention problem slip type; indoor housework—dizziness and leg weakness fall type; and outdoor activities—attention and surface problems fall type.

The number of falls among the elderly varies in different countries, different research populations, and time ranges, which leads to different results. The fall rate of the elderly in this study was 8.7%, which is similar to the results of other studies on the incidence of falls within one year among the elderly in China, such as 10.7% in Shenzhen in one year [41] and 16.6% over two years nationwide in 2018 [31]. A systematic review showed that the fall rate of the Asian population is significantly lower than that of all other ethnic groups, which is 13.89%. The fall rate of the elderly in this study is similar to the fall rate of 7.9–16.2% in European countries in half a year [42]. Studies have shown that the fall injury rate of the elderly in China ranges from 0.6% to 19.5% [7]. In this study, the fall injury rate of the elderly was 6.1%. The relatively low fall rate in this study may be related to the fact that the elderly in this study were all community residents who were generally in good health and did not include hospitalized patients. As shown in a previous study, another possible reason is that elderly Chinese people may hide falls from their families and doctors [43]. Although the retrospective study design can lead to insufficient fall reports, this study reflects the current situation of falls among the elderly residents in our community to some extent.

The musculoskeletal system undergoes physiological changes in the process of aging. Osteoarthrosis can lead to gait instability and limited movement due to muscle atrophy and loose ligaments, which can lead to falls in the elderly [13,24]. The prevalence of osteoporosis and musculoskeletal diseases is higher in women [44]. The variables included both gender and osteoarthrosis, which might be the reason why there is no difference in the rate of falls between different genders in this study, but there is a statistical correlation between osteoarthrosis and falls. Previous studies found that gender differences did not take osteoarthrosis into account [45]. Therefore, in future, we should strengthen osteoarthrosis screening in the elderly, identify high-risk groups, treat and prevent osteoporosis and other diseases, and provide guidance to the elderly who already suffer from osteoarthrosis, to improve the effects of osteoarthrosis on their balance through gait correction and other biomechanical interventions. In addition, a systematic review showed that knee arthropathy was positively associated with falls, while no significant association was found between hip arthropathy and falls [46]. Therefore, the study on the risk of fall caused by different bone and joint injuries can be further explored in the future. This study also found a correlation between poor self-assessed health and falls, which is similar to the conclusions of previous studies, that is, elderly people with poor self-assessed health may have a higher risk of falls.

This study found a correlation between anxiety or depression and falls. Similar results have been found in other studies, with most previous studies having found a significant association between depressive symptoms and falls [15,47,48]. Moreover, a longitudinal study has shown that anxiety symptoms are independent risk factors for fear of falls and activity in the elderly [49], which increases the risk of falls [50]. Although a study showed that the use of antidepressants partially mediates the relationship between depression and falls in the elderly [51], the biological relationship between mental health and falls, such as anxiety and depression, remains to be proven. Therefore, the evidence for the causal relationship between mental health and falls is limited, and more experimental studies should be conducted in the future.

Self-health management awareness in the elderly includes health management skills and knowledge and health behavior development [52], but there is no unified definition of self-health management awareness of the elderly at present. Therefore, there are few studies on the relationship between health management awareness and falls in the elderly. However, in reality, the middle-aged and elderly have a weak sense of self-health management and lack sufficient preventive knowledge [3,19]. Studies have shown that providing health education can effectively reduce the fall rate of the elderly [53]. Therefore, the level of self-health management can be promoted by providing health management measures and community welfare interventions. In experimental studies, interventions to improve the self-management of the elderly have been implemented, such as self-management interventions on exercise, which can reduce the risk of falls [54,55]. This study used health knowledge, health behavior cultivation (diversified diet and regular physical activity), and the utilization of community health service programs to reflect health management awareness. The services of Chinese grass-roots community health service centers include health checkup, health education, and family doctor service. This study found that elderly people who participated in these three basic health services had a lower risk of falls, while the risk for those who did not participate or only participated in one or two basic health services, were not affected. This may be due to the fact that the elderly who participated in all health service activities have a higher awareness of health management and pay more attention to their own health. In addition, it reflects the preventive effect of basic health services on falls. The “Core Health Information for the Elderly” issued by the Ministry of Health of China includes the core knowledge of fall prevention for the elderly [56]. At the national level, China is aware of the seriousness of the fall problem, actively advocating elderly people to prevent falls, including through health services. However, whether the elderly are able to receive these core messages may affect their risk of falls.

Family support plays an important role in the lives of the elderly [46], most of whom live with their children in China. This study revealed that the elderly with a family size of more than two may be more prone to falls, which may be caused by three reasons. First, a more crowded living environment and fewer dedicated family facilities to prevent falls among the elderly. Studies have shown that a total housing area for the elderly larger than 120 square meters lowers the risk of falls [57]. Second, it is also possible that the elderly live with younger generations, and it is very common for grandparents to take care of their grandchildren. Thus, intergenerational relations require older adults to pay more attention to their children and grandchildren and neglect themselves [58,59]. Third, due to the culture of filial piety, some younger generations living with the elderly will do all they can for them, resulting in reduced physical activity and strength in the elderly, increasing the risk of falls [26]. In addition, family members of older people with a history of falls can be overcautious [60], making them more likely to fall again. This study found that spousal support is a protective factor for falls and, similar to previous surveys, having no spouse was a risk factor for falls [61,62]. Therefore, it is necessary to pay more attention to the risk of falls for the elderly who live without a spouse.

In this study, falls were divided into four types according to the combination of location, activity, and cause, which are mainly related to personal and environmental factors. In terms of personal factors, the most common reasons are poor eyesight and lack of concentration. For instance, the elderly are more likely to fall in stairs/hallways and toilets because of poor eyesight and inattention. Often, falls in the elderly are due to visual impairment [63]. In order to reduce the risk factor of poor vision, regular ophthalmic examination should be conducted and appropriate glasses worn. Concurrently, falls at home may be due to lack of light. Thus, the problem of insufficient lighting should be improved [24]. In addition, because of the mentality of not accepting age-related physical changes [64], the elderly might overestimate their physical condition and exercise in a way that is not appropriate for their abilities. It is also possible that indoor housework (dizziness and leg weakness) falls are special cases and might be related to hypotension caused by rapid change of posture and long working hours, as well as poor balance and weak muscles. In terms of environmental factors, the situation of falls due to uneven surfaces is prominent. Among them, the outdoor and toilet fall types are significantly affected by surface problems [65,66]. Among the four types of falls, outdoor falls accounted for the largest proportion, most of which occurred during leisure activities, may be due to the fact that most of the elderly in this study engaged in regular physical activity, were more active, and had the opportunity to engage in leisure activities [24]. The indoor housework fall type accounted for 27.6%, mostly in locations of daily activities, such as the kitchen, living room, and balcony. Moreover, stairs or hallways increase the risk of falls in the elderly and accounted for 27.3% of all falls in this study [67]. Studies have shown that falling on the stairs can lead to the longest hospitalizations in the elderly [68]. This study also found that the falls while taking a bath or going to the toilet accounted for 10.5%, and the bathroom and toilet was one of the common injury locations [67,69]; here, environmental modification is able to reduce the fall rate [70], including adding handrails, increasing the spaciousness of passageways, and increasing lighting. [62,71]. One study showed that the remodeling housing to accommodate the elderly is rare and mainly occurs for people aged 80 and above, due to major diseases and disabilities [71]. However, previous studies have not only explored architectural design for the elderly but also began to describe the spatial walking pattern in daily life [72], the creation of traffic greenways for the elderly [73], and the development of tools to assess environmental safety for the elderly for environmental monitoring [66]. Therefore, China needs to explore more effective measures for environmental transformation, considering the effect of elderly-friendly environmental construction, as well as the experience of fit between the elderly and the environment [74].

This study had some limitations. First, as a cross-sectional study it cannot prove the causal relationship between falls and the factors investigated. In addition, recall bias may have led to the underreporting of falls. Objective indicators, such as balance and gait, were not measured. Therefore, more research is needed on the relationship between physical condition and the risk of falls. Third, the anxiety/depression and physical activity is self-reported, not using professional scales. Thus, there are measurement bias problems in this study. Fourth, factors related to falls analyzed in this study were mainly personal factors, with only a simple descriptive analysis of environmental factors. Thus, there was a lack of analysis of the effect of environmental factors on falls.

## 5. Conclusions

In conclusion, osteoarthropathy, anxiety or depression, self-assessed health status, participation in community health service programs, household size, and spousal support are related to falls. Therefore, society and family members of the elderly should intervene in these influencing factors. In particular, community health education should popularize preventive measures for the elderly according to locations and activities, including the construction of barrier-free environments and the transformation of housing for the elderly should be accelerated in the future in order to reduce the risk of falls in the elderly.

## Figures and Tables

**Figure 1 ijerph-19-14843-f001:**
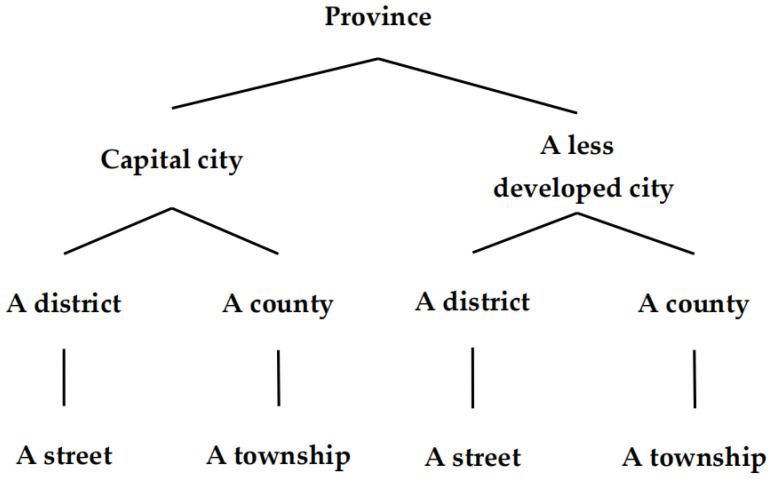
Sampling flow diagram.

**Figure 2 ijerph-19-14843-f002:**
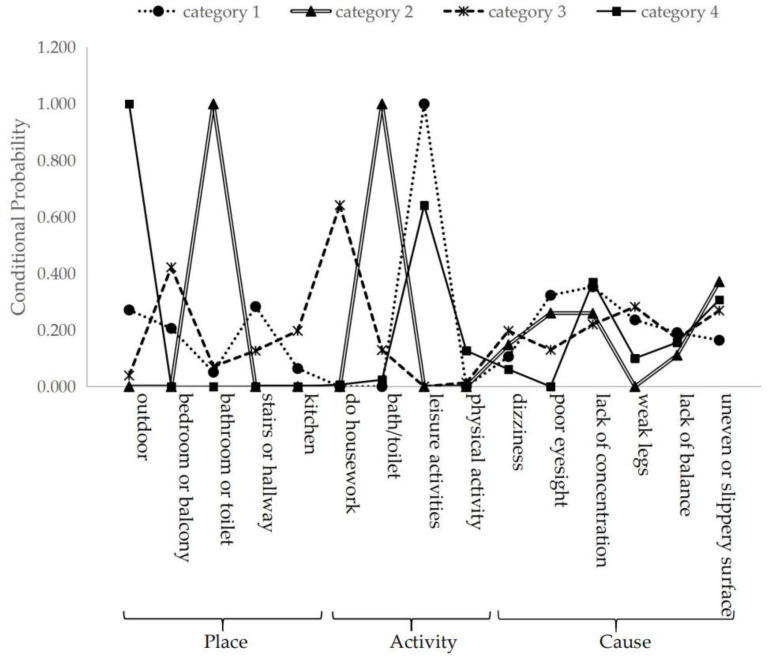
Conditional probability distribution of potential category model.

**Table 1 ijerph-19-14843-t001:** Univariate analysis of falls and injuries in the elderly (*n* = 2994).

Variables		Have Not Fallen (%)	Have Fallen (%)	Statistics (*χ*^2^/*T*)	Total *N*
Gender	Male	1210 (92.1)	104 (7.9)	1.694	1314
	Female	1518 (90.7)	155 (9.3)		1673
Age	<70 age	1623 (91.5)	151 (8.5)	0.130	1774
	≥70 age	1107 (91.1)	108 (8.9)		1215
Household registration	Urban	1345 (91.2)	130 (8.8)	0.088	1475
	Rural	1387 (91.5)	129 (8.5)		1516
Educational level	Illiterate	441 (89.9)	50 (10.2)	5.302	491
	Primary school	897 (92.5)	73 (7.5)		970
	Junior middle school	820 (92.0)	71 (8.0)		891
	Senior high school and above	577 (89.9)	65 (10.1)		642
Monthly income	0–500 CNY	724 (91.1)	71 (8.9)	0.379	795
	500–2000 CNY	1277 (91.2)	123 (8.8)		1400
	2000 CNY	723 (91.9)	64 (8.1)		787
Types of chronic diseases (except osteoarthritis)	None	1234 (92.4)	101 (7.6)	4.256	1335
	1	919 (90.8)	93 (9.2)		1012
	2 or more	573 (89.8)	65 (10.2)		638
Osteoarthritis	No	2435 (92.1)	210 (7.9)	15.927 ***	2645
	Yes	291 (85.6)	49 (14.4)		340
Anxiety/depression	No anxiety/depression	2453 (92.5)	199 (7.5)	34.161 ***	2652
	Anxiety/depression	272 (82.9)	56 (17.1)		328
Self-assessed health status	Better	1368 (89.1)	167 (10.9)	19.701 ***	1535
	Poorer	1365 (93.7)	92 (6.3)		1457
Participation in community health service programs	None	132 (88.0)	18 (12.0)	12.385 **	150
	1 program	590 (91.9)	52 (8.1)		642
	2 programs	903 (89.7)	104 (10.3)		1007
	3 programs	1068 (93.4)	75 (6.6)		1142
Regular physical activity	No	1143 (91.3)	109 (8.7)	0.055	1252
	Yes	1590 (91.5)	147 (8.5)		1737
Health knowledge score		9.79 ± 3.83	9.76 ± 4.07	0.086	9.78 ± 3.85
Spousal support score		4.70 ± 1.84	4.34 ± 1.95	7.649 **	4.67 ± 1.85
Family size	Live alone	240 (92.7)	19 (7.3)	5.415	259
	2 people	1145 (92.6)	92 (7.4)		1237
	>2 people	1313 (90.2)	143 (9.8)		1456
Total	2735 (91.3)	259 (8.7)		2994	

Note: ** *p* < 0.01, *** *p* < 0.001.

**Table 2 ijerph-19-14843-t002:** Multivariate analysis of falls in the elderly (*n* = 2614).

Variables		B	S.E.	Wald Value	*p* Value	OR	95%CI
Osteoarthritis	No	0				1	
	Yes	0.566	0.182	9.716	0.002	1.761	(1.234, 2.513)
Anxiety/depression	No anxiety/depression	0				1	
	Anxiety/depression	0.640	0.180	12.574	<0.001	1.896	(1.331, 2.700)
Self-assessed health status	Better	0				1	
	Poorer	0.390	0.147	7.020	0.008	1.478	(1.107, 1.972)
Participation in community health service programs	None	0		9.738	0.021	1	
	1 program	−0.586	0.300	3.812	0.051	0.556	(0.309, 1.002)
	2 programs	−0.254	0.278	0.834	0.361	0.776	(0.450, 1.337)
	3 programs	−0.651	0.285	5.213	0.022	0.522	(0.298, 0.912)
Family size	Live alone	0		5.437	0.066	1	
	2 people	0.370	0.298	1.542	0.214	1.447	(0.808, 2.594)
	>2 people	0.591	0.281	4.432	0.035	1.806	(1.042, 3.130)
Spousal support score		−0.096	0.039	5.995	0.014	0.909	(0.841, 0.981)
Constant		−2.329	0.367	40.308	<0.001	0.097	

**Table 3 ijerph-19-14843-t003:** Characteristics of falling behavior in the elderly aged 60 years and older (*n* = 259).

Variables		*n*	Percentage (%)
The number of falls	1	211	81.5
	≥2	48	18.5
The type of falls	slip	113	44.5
	trip	72	28.3
	step/sit on an empty	48	18.9
	knocked down	10	3.9
	others	11	4.3
The place of falls	outdoors	104	40.6
	bedroom or living room or balcony	46	18.0
	bathroom or toilet	36	14.1
	stairs or hallway	31	12.1
	kitchen	19	7.4
	others	20	7.8
Activity that occurred when falling	walking or relaxing	98	38.6
	doing housework	46	18.1
	bath/toilet	38	15.0
	leisure activities (except for walking)	30	11.8
	physical activity	11	4.3
	other	31	12.2
The causes of falls	lack of concentration	79	31.1
(multiple choice)	uneven or slippery surface	66	25.5
	weak legs	46	17.8
	lack of balance during exercise	42	16.5
	poor eyesight	41	16.1
	dizziness	31	12.2
	others	14	5.5
Injuries in a fall	uninjured	77	29.7
	epidermal injury	80	30.9
	soft tissue injury	49	18.9
	fracture	47	18.1
	concussion	2	0.8
	other	4	1.5
Total		259	100

**Table 4 ijerph-19-14843-t004:** Fitting test results of each potential category model (*n* = 257).

Category	AIC	BIC	LMR-LRT	BLRT	Entropy	Sample Composition Ratio (%)
Model 1	3533.695	3586.931	-	-	-	100
Model 2	3306.149	3416.171	0.0000	0.0000	0.929	37.4/62.6
Model 3	3181.068	3347.875	0.0000	0.0000	0.996	45.1/40.9/14.0
Model 4	3108.627	3332.219	0.0001	0.0000	0.921	27.3/10.5/27.6/34.6
Model 5	3078.350	3358.727	0.0537	0.0400	0.976	28.8/14.0/23.3/11.3/22.6

## Data Availability

Some or all data are available from the corresponding author upon reasonable request.

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
