# Peer review of "The Incidence of Falls and Related Factors among Chinese Elderly Community Residents in Six Provinces"

_ijerph, 2022, doi:10.3390/ijerph192214843_

Round 1

Reviewer 1 Report

Introduction
"Therefore, large-scale research on falls in the elderly is 50 needed to provide prevention evidence which can be translated into practical recommendations." - I request that the authors use the Corchane database to analyse the scientific evidence on falls and practical recommendations for older people.

Methodology
In the introduction you write that you analysed a group of 60 years and over, while in the methodology and in the abstract you say over 60. The correct limit should be 60 years and over or 65 years and over. Correct this.

Improve the article, especially the part about the methodology of the study, based on the STROBE cheklist. I want to see the different elements of the methodology very carefully and in an orderly way dissected.

2.2 Instrument
How was the survey instrument validated? Has its reproducibility been checked? It is necessary to indicate the scientific basis for setting cut-off points, scores for individual elements e.g. activity level, diet, etc.? Please provide detailed scoring rules for each category.

2.3.2 Physical health
On what basis was the cut-off line established? "In this study, the 152 median was selected as the cut-off value, and scores ≤80 points were defined as poor health, and scores >80 points were defined as good health. Thirdly, anxiety or depression was self-reported." Please provide the method and scientific basis for the calculation.

2.3.3 Self-health Management Awareness
How did the wealth and education of the people surveyed evolve? A favourable financial situation is indispensable for being able to, for example, follow a varied diet. I believe that it is not possible to assess the current situation as self-awareness. Please reformulate this variable and reflect on the objectives of the survey. Also, how, on what basis, was it determined when the diet is correct?
Physical activity should be determined and possibly scored based on WHO guidelines. There needs to be footnotes to the standards on what basis physical activity levels were established and scored.
What is "health education" ? What is this service? Was it freely accessible and available to all respondents?

Results
In my opinion, paradosque results of the type that a varied diet or more family members (which was intended to increase the support of the older person) are associated with more falls may be due to the use of a non-standardised measurement tool. It is necessary to revise the assumptions of the survey, rethink the questions asked and their breakdowns. The list of limitations of the survey and the conclusions that can be drawn from it should be described very carefully.

Author Response

Reviewer1:

  1. Introduction

"Therefore, large-scale research on falls in the elderly is 50 needed to provide prevention evidence which can be translated into practical recommendations." - I request that the authors use the Corchane database to analyse the scientific evidence on falls and practical recommendations for older people.

Response: Thanks for the comments. A systematic review on how to prevent falls in the elderly and summarize the existing research evidence is crucial for our next research work. Therefore, our current study aimed to describe the current status by a large large-scale research in older adults in China and we revised our description in introduction.

  1. Methodology

In the introduction you write that you analysed a group of 60 years and over, while in the methodology and in the abstract you say over 60. The correct limit should be 60 years and over or 65 years and over. Correct this.

Response: Thanks for pointing it out. We have corrected this, the participants of the research were people 60 years old and over.

  1. Improve the article, especially the part about the methodology of the study, based on the STROBE checklist. I want to see the different elements of the methodology very carefully and in an orderly way dissected.

Response: Thanks for the comments. We reorganized and supplemented some details in this part. We have also submitted an attachment file to the STROBE checklist of the methodology.

  1. Instrument

How was the survey instrument validated? Has its reproducibility been checked? It is necessary to indicate the scientific basis for setting cut-off points, scores for individual elements e.g. activity level, diet, etc.? Please provide detailed scoring rules for each category.

Response: Thanks for the comments.

(1) The purpose of original questionnaire was to describe the core health literacy (health knowledge) of the elderly, not specifically designed to study on falls. Therefore, to control the length of questionnaire, some measures of variables were not from scales. However, the questionnaire was designed in the guidance of an experienced expert group, and the content validity could be confirmed. Though no repeated measurement was conducted in the questionnaire pre-experiment, but the factor analysis and reliability analysis were used to test the validity and reliability, and the results were acceptable. We listed these results in methodology. 

(2) Besides spousal support and health knowledge, the variables in our research were measured by 1 item, and the reliability and validity can’t be tested, please see table 1. But the survey instrument is common used in similar research, for example, the self-assessed health was reported on a 0–100 visual analogue scale, which is an accepted instrument in previous studies.

The reliability and validity of the spousal support and health knowledge scores were analyzed. Spousal support: The results of showed that the validity was tested by factor analysis, KMO (Kaiser-Meyer-Olkin) test statistic was 0.695, Bartlett spherical test showed p value <0.001, and one common factor was extracted to explain 80.3% of the variance. In the reliability analysis, Cronbach's alpha was 0.740. It indicates that the reliability and validity of spousal support measurement are acceptable.

Health knowledge scores: The results of showed that the validity was tested by factor analysis, KMO test statistic was 0.871, Bartlett spherical test showed p value <0.001, and one common factor was extracted to explain 43.6% of the variance. In the reliability analysis, Cronbach's alpha was 0.753. It indicates that the reliability of health knowledge measurement was acceptable. The results of validity were not very well, but considering that the survey instrument of health knowledge was designed by the National Department of Aging, the all original 20 items of health knowledge were retained.

(3) The detailed scoring rules for each category have been refined in the manuscript and listed the page in STROBE checklist of the attachment file. (i.e. Physical activity: “For the last six months, do you have regular physical activity?”. (1. Yes 2. No).)

(4) In our research, only “self-assessed health status” have a cut-off value in our research. The selection of the cut-off value is reasonable from the perspective of sample size allocation and the reference of existing research.

  1. Physical health

On what basis was the cut-off line established? "In this study, the median was selected as the cut-off value, and scores ≤80 points were defined as poor health, and scores >80 points were defined as good health. Thirdly, anxiety or depression was self-reported." Please provide the method and scientific basis for the calculation.

Response: Thanks for the comments.

(1) On the one hand, median is the most commonly used statistical method to determine the cut-off value, making the people in each group is more evenly distributed. On the other hand, people who reported a self-assessed health score ≥80 (the median value) had a significantly lower risk of mortality over time relative to people who reported a score <80. (i.e. the following research used median as cut-off value and the cut-off value of self-assessed health score is also 80 point: “Hua X, Lung T W C, Woodward M, et al. Self‐rated health scores predict mortality among people with type 2 diabetes differently across three different country groupings: findings from the ADVANCE and ADVANCE‐ON trials[J]. Diabetic Medicine, 2020, 37(8): 1379-1385.”)

(2) In our research, participants were required to report whether they have depression/anxiety or not, according to their own actual situation. (depression/anxiety: “Please point out your current physical condition based on the actual situation”. (1. No anxiety/depression 2. anxiety/depression) ) Self-reported anxiety and depression have also been widely used. (i.e. “Fountoulakis K N, Apostolidou M K, Atsiova M B, et al. Self-reported changes in anxiety, depression and suicidality during the COVID-19 lockdown in Greece[J]. Journal of affective disorders, 2021, 279: 624-629.”) Since we did not use professional psychological scale, there had certain limitations, which we explained in the limitations.

  1. Self-health Management Awareness

How did the wealth and education of the people surveyed evolve? A favourable financial situation is indispensable for being able to, for example, follow a varied diet. I believe that it is not possible to assess the current situation as self-awareness. Please reformulate this variable and reflect on the objectives of the survey. Also, how, on what basis, was it determined when the diet is correct?

Physical activity should be determined and possibly scored based on WHO guidelines. There needs to be footnotes to the standards on what basis physical activity levels were established and scored.

What is "health education" ? What is this service? Was it freely accessible and available to all respondents?

Response: Thanks for the comments.

(1) We have surveyed the wealth and education status of the people in the research. And In multivariate analysis, wealth and education variables have been included in the model to control. In our study, the diversified diet score shows the basis food categories people have eaten or not, such as vegetables and fruit, which was often used in previous studies. ( “Tao L, Xie Z, Huang T. Dietary diversity and all-cause mortality among Chinese adults aged 65 or older: A community-based cohort study[J]. Asia Pacific Journal of Clinical Nutrition, 2020, 29(1): 152-160.”) However, we finally decided to remove it with the consideration for its suitability in elderly  according to the reviewer’s  suggestion.

(2) In the study, participants self-reported whether they had regular physical activity for last 6 months. (“For the last six months, do you have regular physical activity?”. (1. Yes 2. No).) The definition of physical activity from different research is different, depending on the purpose of each research. Regular physical activity in our research was defined as non-work and housework activities, including but not limited to walking, running, swimming, square dancing, cycling, etc. It is a limitation that we didn't use standardized scale.

(3) In China, essential public health services include health education service are provided to promote the health of community residents. Therefore, health education is accessible and available to all respondents for free.

  1. Results

In my opinion, paradosque results of the type that a varied diet or more family members (which was intended to increase the support of the older person) are associated with more falls may be due to the use of a non-standardised measurement tool. It is necessary to revise the assumptions of the survey, rethink the questions asked and their breakdowns. The list of limitations of the survey and the conclusions that can be drawn from it should be described very carefully.

Response: Thanks for the comments.

(1) Family size is one of variables to describe family’s structure. It has been used in previous studies to measure the family support in some degree though it is not a perfect one. (i.e. “Alcover C M, Chambel M J, Fernández J J, et al. Perceived organizational support‐burnout‐satisfaction relationship in workers with disabilities: The moderation of family support[J]. Scandinavian journal of psychology, 2018, 59(4): 451-461.” & “Xu D, Mou H, Gao J, et al. Quality of life of nursing home residents in mainland China: The role of children and family support[J]. Archives of Gerontology and Geriatrics, 2019, 83: 303-308.”) And some research have also study the relationship of family size and falls among elderly. (i.e. “Kim J, Lee Y S, Kim T H. Effect of number of household members on falls among disabled older people[J]. International journal of environmental research and public health, 2022, 19(10): 5888.”)

We used family size as family support in our study, but the results may not mean that social support doesn’t increase with large family size. It may be that the elderly not receive more attention because of the large family size, or the elderly in large families are more likely to take care of their children or grandchildren. (It was also confirmed in our previous article "Cao W, Yun Q, Chang C, et al. Family support and social support associated with national essential public health services utilization among older migrants in china: a gender perspective[J]. International Journal of Environmental Research and Public Health, 2022, 19(3): 1610.")  And family living space may also be an explanatory factor.

(2) According to the reviewer’s suggestion, we reconsidered the research assumption and finally we removed the variable of diet.

Reviewer 2 Report

Review of a manuscript -Manuscript ID: ijerph-1950453

The paper presents interesting research from six Chinese provinces on the risk of falls in the elderly.

The title encourages you to read the content of the article.

In summary, it is worth emphasizing the purpose of the work.

In my opinion, the purpose of the work has not been precisely specified also in the introduction to the work.

In the material and method section

you can place a participant flow diagram, and instead of text, you can put a table characterizing the participants.

Statistical software for data analysis SPSS 26.0. should be accurately described (manufacturer, company, country, city, year).

The work requires a slight correction.

Author Response

1. In summary, it is worth emphasizing the purpose of the work.

In my opinion, the purpose of the work has not been precisely specified also in the introduction to the work.

Response: Thanks for the comments. In the introduction, we have emphasized the purpose of our research,.

2. In the material and method section

you can place a participant flow diagram, and instead of text, you can put a table characterizing the participants.

Response: Thanks for the comments. We have placed a sampling flow diagram to make the participants involved clearer clear in figure 1. And we have described the characteristics of the participants in table 1.

3. Statistical software for data analysis SPSS 26.0. should be accurately described (manufacturer, company, country, city, year).

Response: Thanks for pointing it out. We have refined the information.